# Orbital disproportionation of electronic density is a universal feature of alkali-doped fullerides

Naoya Iwahara[1] & Liviu F. Chibotaru[1]

Alkali-doped fullerides show a wide range of electronic phases in function of alkali atoms and the degree of doping. Although the presence of strong electron correlations is well established, recent investigations also give evidence for dynamical Jahn–Teller instability in the insulating and the metallic trivalent fullerides. In this work, to reveal the interplay of these interactions in fullerides with even electrons, we address the electronic phase of tetravalent fulleride with accurate many-body calculations within a realistic electronic model including all basic interactions extracted from first principles. We find that the Jahn–Teller instability is always realized in these materials too. In sharp contrast to the correlated metals, tetravalent system displays uncorrelated band-insulating state despite similar interactions present in both fullerides. Our results show that the Jahn–Teller instability and the accompanying orbital disproportionation of electronic density in the degenerate lowest unoccupied molecular orbital band is a universal feature of fullerides.

[1] Theory of Nanomaterials Group, Katholieke Universiteit Leuven, Celestijnenlaan 200F, Heverlee, B-3001 Leuven, Belgium. Correspondence and requests for materials should be addressed to L.F.C. (email: Liviu.Chibotaru@kuleuven.be).

The understanding of electronic phases of alkali-doped fullerides $A_nC_{60}$ is a long-standing and challenging task for material scientists[1]. The prominent feature of these narrow-band molecular materials is the coexistence of strong intra-site Jahn–Teller (JT) effect with strong electron correlation, which underlies the unconventional superconductivity in $A_3C_{60}$ (refs 2–9) and a broad variations of electronic properties in this series of materials in function of the size of alkali ions and the degree of their doping[10–14]. External pressure and insertion of neutral spacers add new possibilities for the engineering of their electronic phases[15–17]. This was recently demonstrated for the $Cs_3C_{60}$ fulleride, which undergoes transitions from Mott–Hubbard (MH) antiferromagnet to a high-temperature superconductor ($T_c = 38\,K$) and then to strongly correlated metal under external pressure[3,4,6–8].

Signs of the JT effect in alkali-doped fullerides were inferred from nuclear magnetic resonance[18,19], infared[20,21], electron energy loss[22,23] spectra, and scanning tunnelling microscopy[24,25] in various compounds. Recently, the parameters governing the complex JT interaction on fullerene anions have been firmly established[26–28], which opened the way for accurate theoretical investigation of the electronic states in fullerides. It was found that in the MH insulating phase of cubic fullerides such as $Cs_3C_{60}$ at ambient pressure, the para-dynamical JT effect is realized as independent pseudorotations of JT deformations at each $C_{60}$ site[29]. The same para dynamical JT effect was found in the metallic phase of $A_3C_{60}$ close to MH transition, whereas the pseudorotation of JT deformation at different sites are expected to be correlated with further departure from the MH transition due to the increase of the band energy[30]. These findings have found confirmation in a very recent investigation of $Cs_3C_{60}$ fulleride, showing an almost unchanged infrared spectrum on both sides in the vicinity of MH metal–insulator transition, whereas displaying its significant variation when the material was brought deeper into the metallic phase[31]. Moreover, our calculations have also shown that the metallic phase in these systems exhibits an orbital disproportionation of electronic density as a result of the dynamical JT instability[30].

This successful theoretical approach is applied here for the investigation of the electronic phase in the $A_4C_{60}$ fullerides, containing an even number of doped electrons per site. We find that these materials exhibit a dynamical JT instability too. As in $A_3C_{60}$, the ground state of $A_4C_{60}$ displays again the orbital disproportionation of electronic density, thus identifying it as a universal key feature of the electronic phases of alkali-doped fullerides.

## Results

**Diagram of JT instability in $A_4C_{60}$.** It is well established that the $t_{1u}$ lowest unoccupied molecular orbital (LUMO) band mainly defines the electronic properties of fullerides[1]. Following the recent treatment of $A_3C_{60}$ (ref. 30), we consider all essential interaction in this band including the one-electron, the bielectronic and the vibronic contributions:

$$\hat{H} = \hat{H}_t + \hat{H}_{bi} + \hat{H}_{JT},$$

$$\hat{H}_t = \sum_{\mathbf{m},\Delta\mathbf{m}} \sum_{\lambda\lambda'\sigma} t^{\Delta\mathbf{m}}_{\lambda\lambda'} \hat{c}^{\dagger}_{\mathbf{m}+\Delta\mathbf{m}\lambda\sigma} \hat{c}_{\mathbf{m}\lambda'\sigma},$$

$$\hat{H}_{bi} = \frac{1}{2} \sum_{\mathbf{m}} \sum_{\lambda\sigma} \left[ U_{\|}\hat{n}_{\mathbf{m}\lambda\sigma}\hat{n}_{\mathbf{m}\lambda-\sigma} + U_{\perp} \sum_{\lambda'(\neq\lambda)\sigma'} \hat{n}_{\mathbf{m}\lambda\sigma}\hat{n}_{\mathbf{m}\lambda'\sigma'} \right.$$
$$\left. - J \sum_{\lambda'(\neq\lambda)} \left( \hat{n}_{\mathbf{m}\lambda\sigma}\hat{n}_{\mathbf{m}\lambda'\sigma} - \hat{c}^{\dagger}_{\mathbf{m}\lambda\sigma}\hat{c}_{\mathbf{m}\lambda'\sigma}\hat{c}^{\dagger}_{\mathbf{m}\lambda-\sigma}\hat{c}_{\mathbf{m}\lambda'-\sigma} - \hat{c}^{\dagger}_{\mathbf{m}\lambda\sigma}\hat{c}_{\mathbf{m}\lambda'\sigma}\hat{c}^{\dagger}_{\mathbf{m}\lambda'-\sigma}\hat{c}_{\mathbf{m}\lambda-\sigma} \right) \right],$$

$$\hat{H}_{JT} = \sum_{\mathbf{m}} \hbar\omega \left[ \sum_{\gamma} \frac{1}{2}\left(p^2_{\mathbf{m}\gamma}+q^2_{\mathbf{m}\gamma}\right) + g\sum_{\lambda\lambda'\sigma}\sum_{\gamma} G^{\gamma}_{\lambda\lambda'}\hat{c}^{\dagger}_{\mathbf{m}\lambda\sigma}\hat{c}_{\mathbf{m}\lambda'\sigma}q_{\mathbf{m}\gamma} \right],$$

$$(1)$$

where, $\mathbf{m}$ denote the fullerene sites, $\Delta\mathbf{m}$ the neighbours of site $\mathbf{m}$, $\lambda,\lambda'$ the $t_{1u}$ LUMO orbitals ($x,y,z$) on each $C_{60}$ (Supplementary Fig. 1), $\sigma,\sigma'$ the spin projections, $\hat{c}_{\mathbf{m}\lambda\sigma}$ and $\hat{c}^{\dagger}_{\mathbf{m}\lambda\sigma}$ are annihilation and creation operators of electron, respectively, $\hat{n}_{\mathbf{m}\lambda\sigma} = \hat{c}^{\dagger}_{\mathbf{m}\lambda\sigma}\hat{c}_{\mathbf{m}\lambda\sigma}$, $q_{\mathbf{m}\gamma}$ and $p_{\mathbf{m}\gamma}$ are the normal vibrational coordinate for the $\gamma$ component of the $h_g$ mode ($\gamma=\theta,\varepsilon,\xi,\eta,\zeta$) and its conjugate momentum, respectively, and $G^{\gamma}_{\lambda\lambda'}$ is Clebsch–Gordan coefficient. The transfer parameters $t^{\Delta\mathbf{m}}_{\lambda\lambda'}$ of $\hat{H}_t$ have been extracted from density functional theory (DFT) calculations (see ref. 30 for $K_3C_{60}$, Methods, Supplementary Methods and Supplementary Table 1 for $K_4C_{60}$). The frequency $\omega$ and the orbital vibronic coupling constant $g$ for an effective single-mode JT model of $C_{60}^{n-}$ have been calculated in ref. 29. The phonon dispersion was neglected, because it is weak in fullerides[1]. The projection of the bielectronic interaction in the $t_{1u}$ LUMO band onto intra-site Hamiltonian ($\hat{H}_{bi}$) is an adequate approximation due to strong molecular character of fullerides[1]. The intra-site repulsion

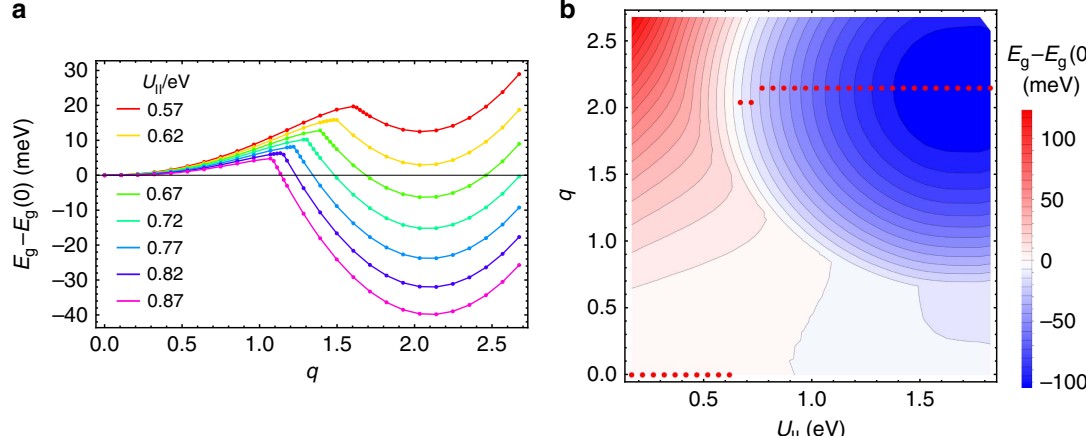

**Figure 1 | Ground-state energy in function of $q$ and $U_{\|}$.** (**a**) Total energy $E_g(q)$ of the ground electronic phase of a $A_4C_{60}$ with cubic band dispersion (see the text) as a function of amplitude of static JT distortion for several values of $U_{\|}$. (**b**) A two-dimensional plot of $E_g(q, U_{\|})$. Red and blue regions stand for positive and negative values proportional to the intensity of the colour. The red points show the amplitude of spontaneous static JT distortion in function of $U_{\|}$. In both figures, the energy at $q = 0$, $E_g(0)$, is subtracted from $E_g(q)$ for each $U_{\|}$.

parameters $U_\parallel$ and $U_\perp$, obeying the relation $U_\parallel - U_\perp = 2J$, are strongly screened: first, by high-energy interband electron excitations reducing their value from 3 eV to $ca$ 1 eV[32] and, second, by intra $t_{1u}$-band excitations. The latter can further reduce $U_\parallel$ and $U_\perp$ several times[32]; however, the extent of this screening strongly depends on the character of the correlated $t_{1u}$ band and can, therefore, be assessed only in a self-consistent manner. On the other hand, the vibronic coupling to the $h_g$ modes, representing a quadrupolar perturbation, is hardly screened. It is the same for the Hund's rule coupling $J$, for which we take the calculated molecular value[29]. We leave $U_\parallel$ as the only free parameter of the theory.

The ground state has been calculated within a self-consistent Gutzwiller approach (see Methods), which proved to be successful for the investigation of $A_3C_{60}$ (ref. 30). To unravel the role played by JT interactions in the ground electronic phase in $A_4C_{60}$, we first consider the case of a face-centred cubic (fcc) $\hat{H}_t$ as in $A_3C_{60}$, the corresponding bands being populated by four electrons per site. Figure 1a shows the calculated total energy as function of the amplitude $q$ of static JT distortions[33,34] of $h_g\theta$ type on fullerene sites. As in the case of $A_3C_{60}$ (ref. 30), the energy curve $E_g(q)$ has two minima, one at the undistorted configuration $q = 0$ and the other at a value $q_0$ approximately corresponding to the equilibrium distortion in an isolated $C_{60}^{4-}$ (see the Supplementary Fig. 2, Supplementary Table 2 and Supplementary Note 1). For $U_\parallel$ smaller than the critical value $U_c \approx 0.64$ eV, the static JT distortion is quenched, $q = 0$. At $U_\parallel > U_c$, the JT distortion reaches its equilibrium value, $q_0$. The full diagram of the total energy $E_g(q, U_\parallel)$ is shown in Fig. 1b (for $E_g$ of $A_3C_{60}$, see Supplementary Fig. 3).

The character of the electronic phase differs drastically in the two domains of $U_\parallel$. The difference is clearly seen in the electron population in the LUMO orbitals $n_\lambda$ and the Gutzwiller's reduction factor $q_{\lambda\lambda}$. The evolution of the population $n_\lambda$ with

respect to $U_\parallel$ (Fig. 2a) shows that for $U_\parallel < U_c$ the phase corresponds to equally populated LUMO bands. This equally populated phase gradually becomes strongly correlated with increasing $U_\parallel$, which is testified by the accompanying decrease of the Gutzwiller's reduction factors for these bands (Fig. 2c). On the contrary, for $U_\parallel > U_c$, it exhibits orbital disproportionation of electronic density among the LUMO orbitals (Fig. 2a) with a sudden jump of the Gutzwiller factor (Fig. 2c).

The existence of the two kinds of phases with and without the JT deformation is explained by the competition between the band energy $\langle \hat{H}_t \rangle$ and the JT stabilization energy in the presence of the strong electron repulsion $U_\parallel$. The former stabilizes the system the most when the splitting of the orbital is absent, whereas the JT effect does by lowering the occupied orbitals. On the other hand, the bielectronic energy is reduced by the quenching of the charge fluctuation (localization of the electrons), which results in the decrease of the band energy and the relative enhancement of the JT stabilization. Therefore, when $U_\parallel$ is small ($U_\parallel < U_c$), the homogeneous (with equal orbital populations) band state is favoured and the JT distortion is quenched. With the increase of $U_\parallel$ over $U_c$, the band energy is reduced to the extent that the JT stabilization on $C_{60}$ sites is favoured, resulting in orbitally disproportionated ground state.

We note that these results are general, which neither depends on the form of the JT distortion on sites nor on the uniformity of these distortions, which can also be dynamical as in $A_3C_{60}$ (ref. 30; *vide infra*).

**Band-insulating state induced by strong electron repulsion**. To better understand the physics of the obtained orbitally disproportionated electronic phase, first consider a simplified model for $\hat{H}_t$, which includes only the diagonal electron transfers after orbital indices, $t_{\lambda\lambda'}^{\Lambda m} = \delta_{\lambda\lambda'} t_{\lambda\lambda'}^{\Lambda m}$ (a widely used approximation

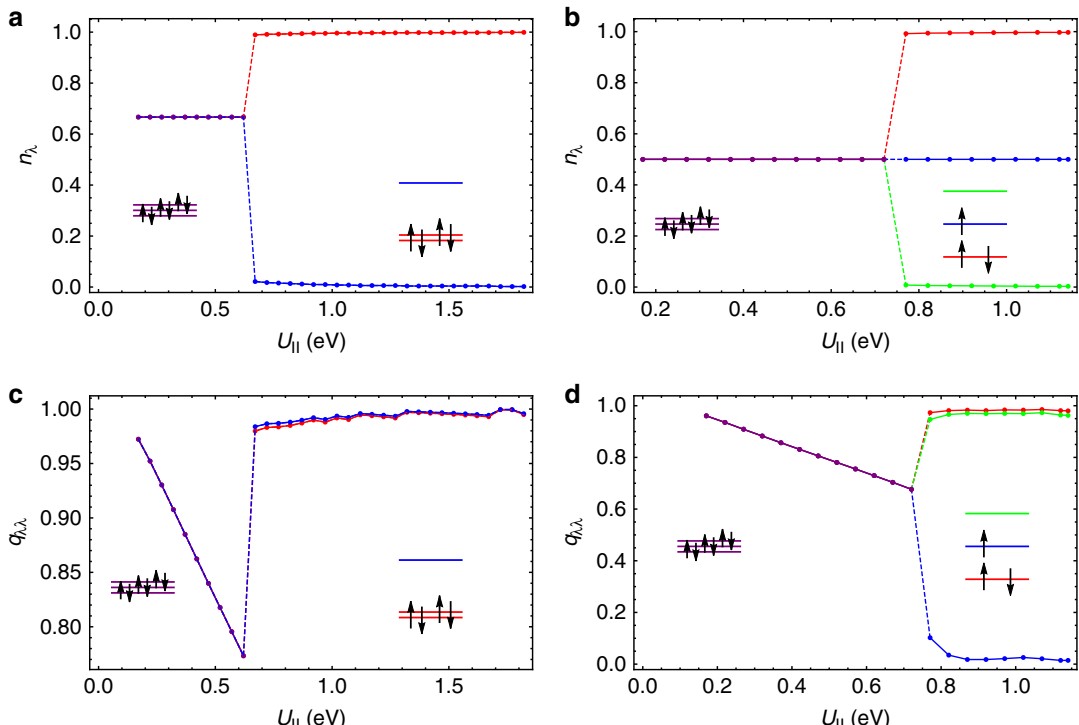

**Figure 2 | Occupation numbers and Gutzwiller reduction factors.** (**a**) Occupation numbers per electron spin of LUMO orbitals $n_\lambda$ and (**c**) Gutzwiller reduction factors in the corresponding bands for a model $A_4C_{60}$ with cubic band dispersion (see the text) subject to static JT interaction as function of $U_\parallel$. (**b,d**) Same as **a** and **c**, respectively, for fcc $A_3C_{60}$.

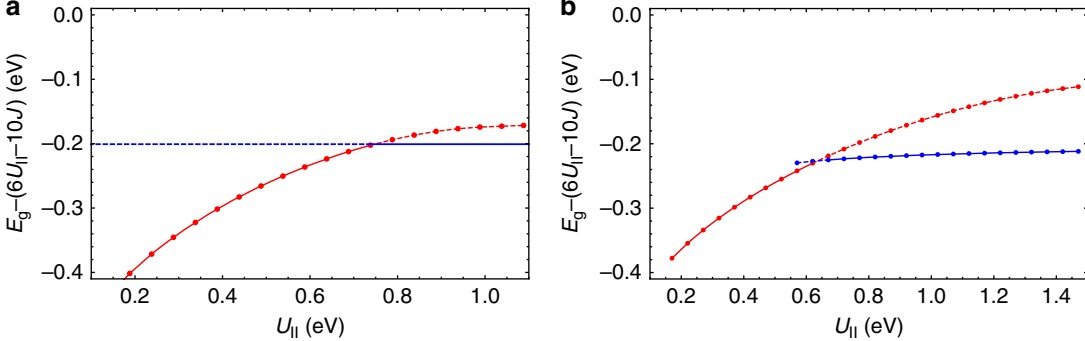

**Figure 3 | Crossing of the correlated metallic and the band-insulating states.** (**a**) Total energy of the ground electronic phase of $A_4C_{60}$ with cubic band dispersion and suppressed interband electron transfer ($t_{\lambda\lambda'}^{\Delta\mathbf{m}} = \delta_{\lambda\lambda'}t_{\lambda\lambda'}^{\Delta\mathbf{m}}$) as function of $U_\parallel$. The red and the blue lines indicate the correlated band solution ($q = 0$) and band-insulating solution with JT splitting, respectively, and the solid and dashed lines indicate the ground and the excited states, respectively, for each $U_\parallel$. The bielectronic energy ($6U_\parallel - 10J$) is subtracted from $E_g$. (**b**) The same for the model of $A_4C_{60}$ with full transfer Hamiltonian used in equation (1).

for the study of multiorbital correlation effects[35–37]). Figure 3a shows the total energies for the two phases with and without JT distortion in function of $U_\parallel$. We see again an evolution of the ground state with the stabilization of orbitally disproportionated electronic phase in the large $U_\parallel$ domain. We find this behaviour pretty similar to the case when the full $\hat{H}_t$ for fcc lattice is considered (Fig. 3b). Owing to the simplification, we can fully identify the orbitally disproportionated phase, because of its exact solution. Indeed, in terms of band solutions $\hat{a}_{\mathbf{k}\alpha\sigma}^\dagger |0\rangle = 1/\sqrt{N}\sum_{\mathbf{m}} e^{i\mathbf{k}\cdot\mathbf{m}}\hat{c}_{\mathbf{m}\alpha\sigma}^\dagger |0\rangle$, where $N$ is the number of sites, we obtain for the orbitally disproportionated phase (see Supplementary Methods):

$$|\Phi_0\rangle = \prod_{\mathbf{k}\sigma}^{\text{all}} \hat{a}_{\mathbf{k}x\sigma}^\dagger \hat{a}_{\mathbf{k}y\sigma}^\dagger |0\rangle, \qquad (2)$$

that is, a pure band state with occupied $x$ and $y$, and empty $z$ band. In the case of a JT distortion different from the $h_g\theta$ type, the solution will be identical to equation (2) but involving band orbitals that are linear combinations of $x$, $y$ and $z$ orbitals. The solution $\Phi_0$ is exact in the whole domain of $U_\parallel$. However, owing to its fully disproportionated character, always corresponding to the orbital populations (2,2,0), it becomes ground state, that is, intersects the correlated homogeneous solution (Fig. 3a), only under the opening of the gap between occupied degenerate orbitals $x$, $y$ and the empty orbital $z$. This means that the orbitally disproportionated phase in Fig. 3a is nothing but conventional band insulator.

The obtained result is not specific to the simplified model. In the case of full $\hat{H}_t$ (Fig. 3b), the orbitally disproportionated state differs only slightly from $\Phi_0$ in equation (2), which is seen from the population of the orbital components of the LUMO band $n_\lambda$ that are close to (2,2,0) (Fig. 2a) and the jump of the Gutzwiller factor to its uncorrelated value 1 (Fig. 2c). Thus, we encounter here a counterintuitive situation: with the increase of the electron repulsion on sites, the system passes from a strongly correlated metal to an uncorrelated band insulator.

To get further insight into the correlated metal to band insulator transition, we compare the electronic state of $A_4C_{60}$ with that of the correlated $A_3C_{60}$, which turns into MH insulator for large $U_\parallel$. In both fullerides, the transition from the orbitally degenerate phase to the disproportionated phase is observed with the increase of $U_\parallel$; however, the nature of the latter phases is significantly different. As orbital disproportionation is indissolubly linked to JT distortions on fullerene sites, either static or

dynamic, the LUMO band in $A_3C_{60}$ will be split in three orbital subbands. Figure 2b,d show that the lowest orbital subband in $A_3C_{60}$ becomes fully occupied and practically uncorrelated ($q_{\lambda\lambda} \approx 1$) with increase of $U_\parallel$ in very close analogy with the behaviour of the two lowest subbands in $A_4C_{60}$ (Fig. 2c). At the same time the electron correlation in the middle half-occupied subband gradually increases, implying that the MH transition basically occurs in this subband[30]. Indeed, the bielectronic energy is reduced by quenching the charge fluctuations in the half-filled middle subband. This is seen as the decrease of the Gutzwiller's factor with the increase of $U_\parallel$ (Fig. 2d), testifying about suppression of the intersite electron hopping. On the contrary, the doubly occupied orbitals are not subject to electron correlation (Gutzwiller's factor becomes close to 1; Fig. 2d). In the case of $A_4C_{60}$, the LUMO orbitals split into two doubly filled orbitals and non-degenerate empty orbital by the JT interaction (see the inset of Fig. 2a). The fully occupied orbitals are similar in nature to those of $A_3C_{60}$, being basically uncorrelated, the same for the empty orbital (all Gutzwiller's factors are close to 1; Fig. 2c).

**Stabilization of orbitally disproportionated phase.** The necessary condition for achieving the band-insulating state is that in the atomic limit of large $U_\parallel$, the orbitally disproportionated molecular state ($S = 0$) has lower energy than the homogeneous $S = 1$ Hund state on each $C_{60}$. Consider the $t_{1u}$ orbital shell of one single fullerene site. Owing to the Hund's rule coupling, the high-spin configurations ($S = 1$), for example, (2,1,1), are stabilized by $3J$ with respect to the low-spin configurations ($S = 0$), for example, (2,2,0). The high-spin (Hund) state always contains half-filled orbitals and leads, therefore, to MH insulator in the limit of large $U_\parallel$. On the other hand, in the presence of a relatively strong static JT effect, the low-spin state is stabilized by $E_{JT} = 4E_{JT}^{(1)}$, where $E_{JT}^{(1)} = \hbar\omega g^2/2$ is the JT stabilization energy in $C_{60}^-$ (refs 33,34). Thus, the low-spin state and, consequently, the band-insulating state are realized as the ground state when the condition $E_{JT} > 3J$ is fulfilled. With the estimate $E_{JT}^{(1)} = 50$ meV and $J = 44$ meV[26,29], we conclude that all $A_4C_{60}$ with hypothetical cubic structure will be band insulators in the static JT limit at sufficiently large $U_\parallel$.

This condition is modified when there is an intrinsic orbital gap $\Delta_0$ at fullerene sites, which arises due to the lowering of the symmetry of the crystal field (CF) in non-cubic fullerides (Table 1). Band structure calculations of $A_4C_{60}$ with body-centred

**Table 1 | Criterion for the transition from correlated metal to band insulator.**

| | Intrinsic orbital splitting | Static JTE | Dynamical JTE | Band insulator |
|---|---|---|---|---|
| (A) | $\equiv$ | − | − | Never |
| | | + | − | $E_{JT} > 3J$ |
| | | + | + | $E_{JT} + \hbar\bar\omega > 3J$ |
| (B) | $\Delta_0$ | − | − | $\Delta_0 > 3J$ |
| | | + | − | $\Delta_0 + E_{JT} > 3J$ |
| | | + | + | $\Delta_0 + E_{JT} + \frac{1}{2}\hbar\bar\omega > 3J$ |
| (C) | $\Delta_0$ | − | − | Never |
| | | + | − | $E_{JT} > 3J$ |
| | | + | + | $E_{JT} + \frac{1}{2}\hbar\bar\omega > 3J$ |
| (D) | $(1-\mu)\Delta_0$ ; $\mu\Delta_0$ | − | − | $(1-\mu)\Delta_0 > 3J$ |
| | | + | − | $(1-\mu)\Delta_0 + E_{JT} > 3J$ |
| | | + | + | $(1-\mu)\Delta_0 + E_{JT} > 3J$ |

+ / −, presence/absence; $\Delta_0$, the (non-JT) crystal-field splitting of the $t_{1u}$ LUMO shell on one fullerene site; $0 \leq \mu \leq 1$; $E_{JT}$, JT stabilization energy for $C_{60}^{4-}$; $\hbar\bar\omega/2$, energy gain due to the JT dynamics per dimension of the trough; JTE, Jahn–Teller effect.
The criterion for correlated metal to band insulator transition in threefold degenerate band system with four electrons per site*.
*The Hund's rule energy 3 J will be slightly modified by taking into account the multiplet structure due to the presence of two low-spin terms in $Cs_{60}^{4-}$.

tetragonal (bct) lattice (Supplementary Fig. 4) show that the low-symmetry CF is weak and does not admix the excited electronic states on fullerene sites. Accordingly, the strength of the JT coupling is not modified by this CF splitting. When one of the $t_{1u}$ orbitals is destabilized by the CF splitting $\Delta_0$ (Table 1, B), the Hund configuration (2,1,1), with $S = 1$, is also destabilized by $\Delta_0$, whereas the energy of the low-spin configuration (2,2,0), with $S = 0$, remains unchanged, because the destabilized orbital is not populated ($n = 0$). The orbitally disproportionated state becomes the ground one when $E_{JT} + \Delta_0 > 3J$, which means that the low-symmetry CF splitting enhances the tendency towards disproportion. Moreover, if the CF splitting $\Delta_0$ is larger than the Hund's rule energy 3J, the system becomes band insulator for sufficiently large $U_{\parallel}$ even in the absence of the JT effect ($E_{JT} = 0$).

On the contrary, if two $t_{1u}$ orbitals are equally destabilized by $\Delta_0$ (Table 1,C), both the high-spin and the low-spin configurations are destabilized by $2\Delta_0$; thus, the system does never become band insulator only due to CF splitting. The band insulator is achieved in this case only when the JT stabilization in the low-spin state is stronger than the Hund energy 3J, which results in the same criterion as for the degenerate case (Table 1, A). We stress that the amplitude of the CF splitting does not play a role in this case. It only plays a role when the destabilizations of the low- and high-spin configurations are different, such as in the case of the second scenario (Table 1,B) or the last one (Table 1,D) corresponding to complete CF lift of degeneracy. In the latter case, on the argument given above, only the CF splitting between the highest two orbitals adds to the criterion, which looks now as intermediate ($0 < 1 - \mu < 1$, see Table 1) to the previous scenarios, (Table 1, B and C).

According to the tight-binding simulations of the DFT LUMO band (Fig. 4a), the pattern of the orbital splitting for the bct $K_4C_{60}$ corresponds to the third scenario of the CF splitting (Table 1, C) with a gap $\Delta_0$ of ca. 130 meV. Given a similar lattice structure, the same situation is expected also for $Rb_4C_{60}$. Therefore, according to the criterion in Table 1, no band-insulating state can arise in these two fullerides, unless the JT stabilization energy exceeds the Hund energy (3J). Following the estimations of $E_{JT}^{(1)}$ and $J$ (see above), we conclude that the uncorrelated band-insulating phase is stabilized in $A_4C_{60}$ with $A = K$, Rb, in agreement with experiment. In body-centred orthorhombic (bco) $Cs_4C_{60}$, the low-symmetric CF will completely lift the degeneracy of the $t_{1u}$ orbitals, leading to a scenario D in Table 1. The splitting between the highest and the middle $t_{1u}$ orbitals will enhance the tendency towards the stabilization of the band-insulating state, according to the criterion in Table 1.

Finally, we consider the effect of the JT dynamics on the stabilization of the orbitally disproportionated phase. In the cubic $A_4C_{60}$, due to a perfect disproportionation (2,2,0) of the occupation of orbital subbands, the dynamical JT effect on the fullerene sites will be unhindered by hybridization of orbitals between sites pretty much as in metallic $A_3C_{60}$ close to MH transition[30]. The pseudorotation of JT deformations in the trough of the ground adiabatic potential surface of fullerene anion gives a gain in nuclear kinetic energy of $\hbar\bar\omega/2 \approx 30$ meV per dimension of the trough[29]. The gain amounts to $\hbar\bar\omega$ in the case of two-dimensional trough in $C_{60}^{4-}$ (ref. 33,34). This will enhance the criterion for band insulator by $\hbar\bar\omega$ in the case of cubic lattice (Table 1). For relatively large intrinsic CF gap, $\Delta_0 > \hbar\bar\omega/2$, one of the rotational degrees of freedom in the trough will be quenched and the JT dynamics will reduce to a one-dimensional pseudorotation of JT deformations entraining only the two degenerate orbitals in the scenarios of splitting shown in Table 1, B and C. This is apparently the case of bct $K_4C_{60}$ and $Rb_4C_{60}$ at ambient pressure. In the case of last scenario (Table 1, D) of CF splitting, the JT pseudorotational dynamics will be completely quenched if the separations between the three orbitals exceed much $\hbar\bar\omega/2$. Whether this is the case of $Cs_4C_{60}$ with a relevant bco lattice remains to be answered by a DFT-based analysis similar to one done here for $K_4C_{60}$ (Fig. 4a,c).

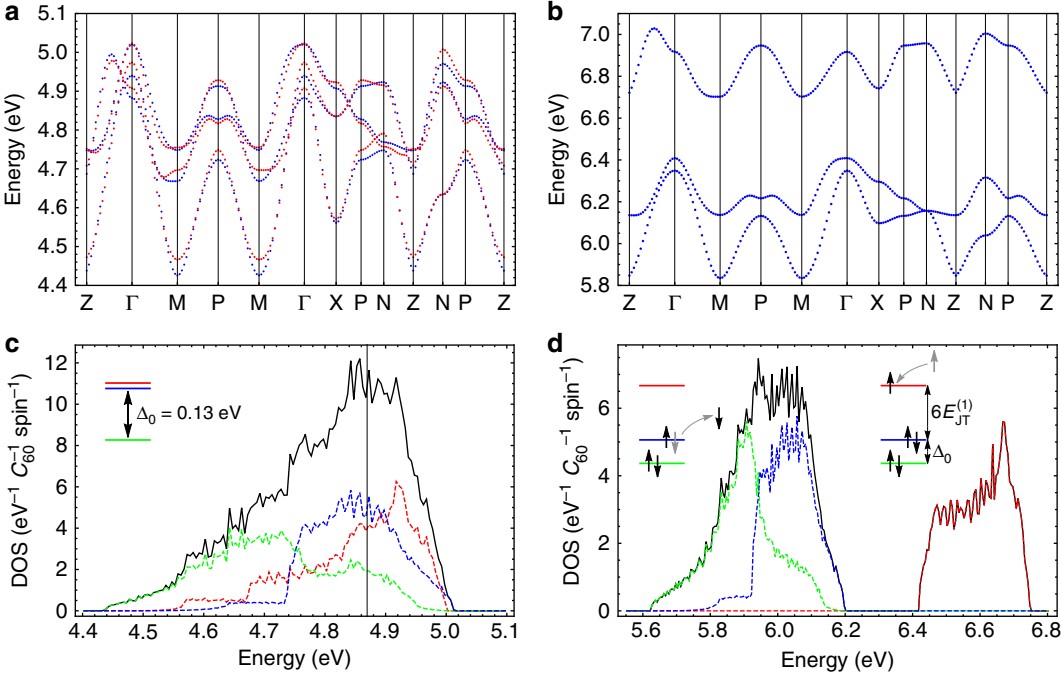

**Figure 4 | LUMO band and density of states.** (**a**) LUMO band dispersion and (**c**) corresponding density of states of $K_4C_{60}$ calculated by DFT (generalized gradient approximation) for experimental structure. (**b**) Dispersion of single-particle excitations and (**d**) the corresponding density of states corresponding to the orbitally disproportionated ground state at $U_{||} = 0.5$ eV and without hybridization between the occupied and empty band orbitals. The blue dots in **a** and **c** show the tight-binding simulation, and red dots in (**a**) the DFT calculations. The black line in **c** and **d** corresponds to a full DOS, whereas the coloured lines the orbitally projected DOS.

Another ingredient defining the transition from the correlated metal to band insulator is the bielectronic interaction $U_{||}$. The value of $U_{||}$ at which the band-insulating state is stabilized (the crossing point of the two phases in Fig. 3) depends on the relation between the band energy in the homogeneous correlated metal phase $\langle \hat{H}_t \rangle$ and the gain of intra-site energy due to disproportionated orbital occupations (static and dynamic JT stabilization energies). The calculations (Fig. 3) show that in the cubic model of $A_4C_{60}$, the band-insulating state arises already at modest values of $U_{||}$, which means that it is always achieved in these fullerides (cf. experimental Hubbard $U \approx 0.4$–$0.6$ eV for $K_3C_{60}$ (refs 38,39)). As the necessary conditions for the cubic and bct $A_4C_{60}$ are the same (Table 1), the band-insulating state seems to be well achieved in the bct $K_4C_{60}$ and $Rb_4C_{60}$. The stabilization of the band-insulating state in the bco $Cs_4C_{60}$ seems to be facilitated by a larger $U_{||}$ expected due to the larger distance between $C_{60}$ sites. This is in line with the experimental observation of insulating non-magnetic state in all $A_4C_{60}$ at ambient pressure[13,14,40].

We want to emphasize that the intrinsic CF splitting of the $t_{1u}$ LUMO orbitals on $C_{60}$ sites in fullerides does not render them automatically band insulators. Thus, the DFT calculations of $K_4C_{60}$ (Fig. 4a,c) do not give a band insulator, but rather a metal despite the intrinsic CF splitting of 130 meV (see Supplementary Fig. 5 for Brillouin zone). The same situation is realized in $Cs_4C_{60}$ and any other fulleride in which the intrinsic CF splitting is significantly smaller than the uncorrelated bandwidth. The band-insulating state (Fig. 4b,d) only arises due to JT distortions on fullerene sites and due to the effects of electron repulsion in the $t_{1u}$ shell reducing much the band energy of the homogeneous metallic state.

In general, the band-insulating state will be achieved at any value of the gap between the highest and the middle LUMO orbitals $\Delta$ (a sum of CF and JT splittings) at $C_{60}$ sites, which fulfills the necessary condition in Table 1. The only difference is that smaller $\Delta$ will require larger $U_{||}$ for achieving the intersection with the homogeneous correlated metal phase (Fig. 5). One should note that the band-insulating state arises not only three-orbital systems such as fullerides, but also in other orbitally degenerate systems with even numbers of electrons per site when both $\Delta$ and $U_{||}$ are sufficiently large. Thus, the scenario B without JT effect in Table 1 was considered for a one-third-filled three-orbital model with infinite-dimensional Bethe lattice[37].

**Universality of orbital disproportionation in fullerides.** Given the established orbital disproportionation of the LUMO electronic density in $A_3C_{60}$ (refs 29,30), its persistence in $A_4C_{60}$ found in the present work makes the orbital disproportionation a universal feature of electronic phases in alkali-doped fullerides. Indeed, the same electronic phase is expected also for $A_2C_{60}$ fullerides[13,18], which are described by essentially the same interactions as $A_4C_{60}$. The only difference will be the inversion of the intrinsic CF and JT orbital splittings on the fullerene sites.

The existence of the orbital disproportionation in fullerides is imprinted on their basic electronic properties. As discussed in 'Band-insulating state induced by strong electron repulsion' and ref. 30, in the disproportionated phase of metallic $A_3C_{60}$ the orbital degeneracy is lifted and the electron correlation develops in the middle subband, whereas it does not play a role in other subbands. Therefore, the MH transition also mainly develops in the middle subband[30] and, hence, one has no ground whatsoever to claim strong effects of orbital degeneracy on the MH transition in these materials as was done repeatedly in the past[35,41,42]. Another important manifestation of the orbital

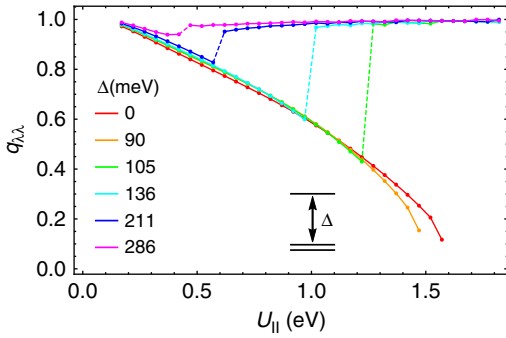

**Figure 5 | Transition from correlated metal to band insulator induced by orbital splitting.** Evolution of the the Gutzwiller reduction factors $q_{\lambda\lambda}$ for $A_4C_{60}$ within the cubic model used in Fig. 1 in function of $U_{\parallel}$ for different orbital gaps $\Delta$, which are sums of JT and CF splittings (the former is considered arbitrary now). The monotonously decreasing line corresponds to a correlated metal, which for $\Delta < \Delta_c$ ($\Delta_c \approx 100$) evolves into a MH insulator. The jumps to $q_{\lambda\lambda} \approx 1$ for values $\Delta > \Delta_c$ correspond to onsets of band insulator.

disproportionation is the similar JT dynamics corresponding to independent pseudorotation of JT deformations on different fullerene sites in both MH phase[29] and strongly correlated metallic phase[30] of $A_3C_{60}$. This has recently found a firm experimental confirmation in the equivalence of infrared spectra of the corresponding materials[31].

In $A_4C_{60}$, the experimental evidence for the (2,2,0) orbital disproportionated phase comes, first of all, from the observed non-magnetic insulating ground state. Moreover, as implied by the intersection picture of the two ground phases (Fig. 3), the correlated metal to band insulator transition could be observed by the decrease of the bielectronic interaction $U_{\parallel}$ with respect to the band energy. This seems to be realized as the metal–insulator transition in $Rb_4C_{60}$ under pressure[43], where the electron transfer (band energy) is enhanced by the decrease of the distance between the sites and $U_{\parallel}$ is concomitantly reduced by the enhanced screening.

Further evidence for the orbitally disproportionated phase comes from spectroscopy. In the case of static JT distortions of $h_g\theta$ type on fullerene sites, the single-particle excitations are exactly described by the uncorrelated band solutions, $|\Phi^e_{z\mathbf{k}\sigma}\rangle = \hat{a}^\dagger_{z\mathbf{k}\sigma}|\Phi_0\rangle$ for electron and $|\Phi^h_{\alpha\mathbf{k}\sigma}\rangle = \hat{a}_{\alpha\mathbf{k}\sigma}|\Phi_0\rangle$, $\alpha = x, y$, for hole quasiparticles, respectively (see Supplementary Note 2). Figure 4 shows that the dispersion of electron- and hole-like excitation basically corresponds to the decoupled $z$ and $(x, y)$ bands due to practically suppressed hybridization between occupied and unoccupied LUMO orbitals (Supplementary Figs 6 and 7) when the band gap opens. The hole-like excitations (Fig. 4d) show the density of states closely resembling the width and the shape of the LUMO feature in the photoemiossion spectrum[44].

## Discussion

In this study, we investigated theoretically the ground electronic phase of $A_4C_{60}$ fullerides. It is found that the relatively strong electron repulsion on $C_{60}$ sites stabilizes the uncorrelated band-insulating state in these materials. A particular conclusion of the present study is that the widely used term 'Jahn–Teller–Mott insulator'[20,23,45,46] is not appropriate here, because it involves mutually excluding phenomena. $A_4C_{60}$ or any similar multi-orbital system with even number of electrons per sites can be either a correlated metal with no JT distortions, high-spin (Hund) MH insulator, or uncorrelated band insulator stabilized

by static or dynamic JT distortions. We prove here that the latter is the case in the fullerides due to a weaker Hund's rule interaction compared with JT stabilization energy, which is ultimately due to relatively large radius of $C_{60}$. Similar situation should arise in other crystals with large unit cells with local orbital degeneracy, the first candidate being the molecular crystals of $K_4$ clusters[47]. The present demonstration of the persistence of band-insulating phase in $A_nC_{60}$ with even $n$ identifies the orbital disproportionation of the LUMO electronic density as a universal key feature of all alkali-doped fullerides, which undoubtly has a strong effect on their electronic properties. We would like to emphasize that the ultimate reason of orbital disproportionation in fullerides is the existence of equilibrium JT distortions, static or dynamic, on fullerene sites. These are always present in fullerides due to the crucial effect of electron correlation on the JT instability of $C_{60}^{n-}$ sites.

## Methods

**Self-consistent Gutzwiller's approach.** The ground states of $A_4C_{60}$ were calculated using the self-consistent Gutzwiller's approach developed for the JT system[30]. Within this approach, both the JT effect and the electron correlation are simultaneously treated by introducing the orbital specific Gutzwiller's variational parameter in the Gutzwiller's wave function, $|\Psi_G\rangle = \hat{P}_G|\Phi_S\rangle$, where $\Phi_S$ is a Slater determinant and $\hat{P}_G$ is the Gutzwiller's projector. Besides the Gutzwiller's parameters in $\hat{P}_G$, the orbital coefficients in the Slater determinant $\Phi_S$ are also treated as variational parameters. The total energy was minimized with respect to both Gutzwiller's parameter and the orbital coefficients self-consistently (see Supplementary Methods and ref. 30 for detail).

**DFT calculations.** The transfer parameters $t^{\Delta\mathbf{m}}_{\lambda\lambda'}$ were taken from ref. 30 for fcc $K_3C_{60}$ and derived from the DFT calculations for bct $K_4C_{60}$. The DFT calculations were performed within the generalized gradient approximation with the pseudopotentials C.pbe-mt_fhi.UPF and K.pbe-mt_fhi.UPF of QUANTUM ESPRESSO 5.1 (ref. 48). The nuclear positions were relaxed, whereas the lattice constants from ref. 20 were fixed. The tight-binding parameters were obtained by fitting the DFT band to the model transfer Hamiltonian ($\hat{H}_t$) including the nearest-neighbour and next nearest-neighbour terms. The results are shown in Fig. 4a. For the model transfer Hamiltonian and the obtained 18 parameters, see the Supplementary Methods and Supplementary Table 1, respectively.

**Data availability.** The data in this manuscript are available from the authors on request.

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

## Acknowledgements

We acknowledge useful discussions with Dennis Arčon, Katalin Kamarás, Erio Tosatti and Martin Knupfer. N.I. is an overseas researcher under Postdoctoral fellowship of Japan Society for the Promotion of Science. N.I. also acknowledge the financial support from the Flemish Science Foundation (FWO) and the GOA grant from KU Leuven.

## Author contributions

N.I. made the calculations. L.F.C. conveived the idea and guided the work. Both authors discussed the results and wrote the manuscript.

## Additional information

**Competing financial interests:** The authors declare no competing financial interests.

