## [Peer review file · Nature Communications]

Reviewers' comments:

Reviewer #1 (Remarks to the Author):

Alkali-doped fullerides have long attracted interest for their superconducting nature and possible electronic correlation effects coming from the energy competition between the narrow degenerate electronic bands (three t_{1u} bands) and hard optical phonons. Numerous model analyses have been carried out for possible novel phenomenon emerging from this competition. The work by Iwahara and Chibotaru is among such theoretical analyses and addresses a phenomenon of "orbital disproportionation"--an imbalance of the electronic occupation between the t_{1u} bands--derived from the electron-phonon coupled Hamiltonian with a modern numerical solver for strong correlation. They have found this phenomenon before in A3C60 with the same scheme as the present one. The novelty of the present work that the authors want to expose is in their finding of the same phenomenon in A4C60 and discussion of its generality to AnC60 ($n=2-4$).

I believe that their work is an interesting contribution to the field. What compromises the value of this manuscript, with which I am reluctant to recommend the publication with the present form, is the lack of HOW and WHY that orbital-disproportionation state is generally stabilized. Only "unexpected (second paragraph in Sec. IIA)" and "counterintuitive (First paragraph in Sec.IIB)" numerical results are not persuasive for general readers to accept that the authors grasp the reality since this is not an experimental work; the numerical ground state of the electronic model with strong interactions is basically sensitively dependent on how to construct and solve the model. With plausible analysis and/or explanation on (i) how the onsite U_{\parallel} promotes the disproportionation and (ii) why it applies to general electronic occupation n , I can reconsider the possible publication of the manuscript in Nature Communications.

Actually I have a speculation about these two points. Namely, the "negative J " mechanism for singlet superconductivity (for A3C60) by Capone [Science 296, 2364 (2002); PRL 93, 047001 (2004); for example] may apply. In addition to the electronic J , there is also a contribution mediated by phonons having the opposite sign. If the latter contribution dominates, U_{\perp} become larger than U_{\parallel} via $U_{\parallel}-U_{\perp}=2J$, which obviously promotes the double occupation (which always corresponds to the intra-orbital singlet configuration). A recent work by Nomura [Sci. Adv. 1, e1500568 (2015)] has shown that this is really the case in A3C60 with the first-principles estimate of the effective interactions U_{\parallel} , U_{\perp} and J integrating out the phonon degrees of freedom. Since this estimation excludes the screening processes within the t_{1u} bands, it seems to apply to general occupation n . The role of U_{\parallel} has been also simply explained that it renormalizes the band to make the difference $U_{\parallel}-U_{\perp}$ relevant compared with the kinetic energy. Although their theory concerns the dynamical-type Jahn-Teller effect, similar discussion in terms of the effective interactions could be given for the present static-JT case.

Of course this scenario is just a speculation and so different interpretations are, if possible, welcomed. Anyway, convincing discussions about the origin of the general stability of the disproportionation state is essential. Fortunately, the disproportionation state is uncorrelated according to the authors; it should allow simple explanations (e.g., discussion in terms of effective interactions based on the mean-field picture).

Reviewer #2 (Remarks to the Author):

The present manuscript describes the results of theoretical calculations of the electronic phase diagram of A4C60, an alkali-doped fulleride with an even doping number. The results presented here are an extension of the methodology developed by the same authors and applied to the metallic and insulating phases of A3C60. The authors' work revolves around the importance of the

dynamic/static Jahn-Teller effect in contributing towards the rationalization of the electronic properties of fulleride solids. Overall I find the ideas and results here of interest but the manuscript is written in a somewhat opaque and cryptic way making it hard to follow the development of the results. Definitely it should be re-written in a more logical and coherent way to allow the reader to follow the results.

The major message of this work is the universality of orbital disproportionation in fulleride physics. But unlike A3C60 which is cubic (face-centered or body-centered-derived), all known A4C60 fullerides suffer from a reduction of symmetry to either tetragonal (K4C60, Rb4C60) or orthorhombic (Cs4C60). It is unclear to the present referee what the role of orbital disproportionation is here when the degeneracy of the t_{1u} orbitals is lifted. In the case of Cs4C60 (which is not considered by the authors) there is full lifting of the degeneracy.

The authors begin by using a hypothetical cubic fcc structure for A4C60 which could be a starting model of course. They do say that this does not affect the main conclusions but I really cannot see that this is developed in the manuscript. There is Table I where the last line should correspond to "real" materials. Here the crystal field has already partially removed the degeneracy but its magnitude does not enter the criteria definitions. For the parameters used, the material should be always a band insulator rendering the conclusions entirely trivial. Moreover body-centered orthorhombic Cs4C60 will have three non-degenerate states rendering the relevance of the JT effect questionable.

I would really like to see the treatment by the authors presented in a coherent way. One can start by the fcc model but if one wants to address real systems one should emphasize the treatment of the bct and bco structures. While the splitting of the levels in experimental bct (because $c < a$) is like shown in Fig 5c inset, most of the other figures deal with inverted case relevant to $a < c$ (e.g. Fig 4). In addition, various parameters appear without much explanation, e.g. we can see Δ , Δ_0 , Δ_c .

It is not clear to me what the authors imply by saying that A4C60 are uncorrelated band-insulators despite the large on-site repulsions. It looks like one is playing with words here. What is the physical significance of such statements.

I find interesting the statement that the picture developed before mainly by the Tosatti/Gunnarsson groups of the importance of degeneracy in stabilizing the highly-correlated metallic state in A3C60 is inappropriate. To this effect they stress the importance of the link between the JT effect and the lifting of the degeneracy. This is a very interesting issue. On the other hand, it will be fair to also refer to theoretical work which does not include any phonon contributions, notably the recent work by Kivelson which finds that a low-spin ground state (ie orbital degeneracy lifted) can occur purely on electronic grounds in fullerenes (PRB 93, 165406 (2016)). All these are interesting ideas in this exciting field and one needs to keep an open mind.

I think this paper is possibly publishable but the authors should improve its readability considerably and give a more convincing explanation of the relevance of "orbital disproportionation" for systems which are never cubic (cf. A3C60) and for which in reality there is always a strong competing crystal field.

Reviewer #3 (Remarks to the Author):

In this work the authors present a rather detailed analysis of a quite cumbersome model, containing several phenomenological parameters to describe both the electron correlation and the Jahn-Teller effect.

The main effect is that the Coulomb repulsion, since provides an effective repulsion between

empty and occupied states, can amplify the Jahn-Teller distortion and provide a change in the occupied states, leading to band insulator behavior.

This effect is qualitatively reproduced within the simple Hartree-Fock theory (see e.g. PRB 62, 7619) that, as expected, overestimates the gap.

The present work represents an improvement of previous works, since the Gutzwiller approximation may be more quantitative than the simplest Hartree-Fock theory.

However I do not think this paper deserves to be published in this Journal because of lack of importance and novelty of the results presented.

Moreover from the experimental point of view, if a critical value of U is necessary for a Jahn-Teller orbital disproportionation of electronic density, this should be a measurable effect (e.g. by increasing the temperature).

However I do not see any evidence from the experimental data, and the model and approximations used, certainly require some experimental confirmation.

For the above reasons, though the paper could be published in some form, I do not recommend for publication in this Journal.

Response to the reviewers' comments

First of all, we would like to thank all reviewers for valuable comments which helped us to improve our manuscript. In the revised version of the manuscript we took into account all their comments and suggestions.

Reviewer #1

Comment #1:

I believe that their work is an interesting contribution to the field. What compromises the value of this manuscript, with which I am reluctant to recommend the publication with the present form, is the lack of HOW and WHY that orbital-disproportionation state is generally stabilized. Only "unexpected (second paragraph in Sec. IIA)" and "counterintuitive (First paragraph in Sec.IIB)" numerical results are not persuasive for general readers to accept that the authors grasp the reality since this is not an experimental work; the numerical ground state of the electronic model with strong interactions is basically sensitively dependent on how to construct and solve the model.

Our response:

In the revised version of the manuscript we added three paragraphs (end of Sect. IIA and IIB, beginning of Sect. IIC) explaining in detail the origin and stabilization of the two competing phases in fullerenes, the homogeneous correlated metallic phase and the non-correlated band-insulating phase. It can be seen that the nature and stabilization of these phases becomes physically clear without numerical calculations. In particular, we demonstrate that the band-insulating state is an exact solution of the Hamiltonian (1) without resort to any numerical calculations.

We agree with the Reviewer that “unexpected” is inappropriate term for the description of the existence of the two electronic phases in fullerenes. However, the term “counterintuitive” is justified since the uncorrelated band-insulator solution becomes stabilized only for sufficient large Hubbard U , the increase of which is generally expected to enhance the correlation effects.

Comment #2:

With plausible analysis and/or explanation on (i) how the onsite U_{\parallel} promotes the disproportionation and (ii) why it applies to general electronic occupation n , I can reconsider the possible publication of the manuscript in Nature Communications.

Our response:

The added detailed analysis on the two electronic phases in the revised version of the manuscript already answer this comment. Here we summarize the main point:

(i) The homogeneous (with equally populated t_{1u} LUMO orbitals) correlated metallic phase maximizes the gain of the band energy, whereas the orbitally disproportionated phase maximizes the gain of Jahn-Teller stabilization energy and of the crystal-field splitting energy (in non-cubic lattices) on the C_{60} sites. With the increase of U_{\parallel} , the band energy is reduced in the former phase to a point that its gain become smaller than the gain of Jahn-Teller stabilization energy (and crystal-field splitting energy if any) at values exceeding the critical one, $U > U_c$. At this moment the orbitally disproportionated phase (corresponding a band insulator state) becomes the ground one.

(ii) Given that the competition between the band energy and the intrasite electron stabilization energy is similar in both A_3C_{60} and A_4C_{60} (A_2C_{60}), i.e. in fullerides with both even and odd numbers of electrons per C_{60} site, the orbitally disproportionated states arise at any n for sufficiently large U .

Comment #3:

Actually I have a speculation about these two points. Namely, the "negative J " mechanism for singlet superconductivity (for A_3C_{60}) by Capone [Science 296, 2364 (2002); PRL 93, 047001 (2004); for example] may apply. In addition to the electronic J , there is also a contribution mediated by phonons having the opposite sign. If the latter contribution dominates, U_{\square} become larger than U_{\parallel} via $U_{\parallel} - U_{\square} = 2J$, which obviously promotes the double occupation (which always corresponds to the intra-orbital singlet configuration). A recent work by Nomura [Sci. Adv. 1, e1500568 (2015)] has shown that this is really the case in A_3C_{60} with the first-principles estimate of the effective interactions U_{\parallel} , U_{\square} and J integrating out the phonon degrees of freedom. Since this estimation excludes the screening processes within the t_{1u} bands, it seems to apply to general occupation n . The role of U_{\parallel} has been also simply explained that it renormalizes the band to make the difference $U_{\parallel} - U_{\square}$ relevant compared with the kinetic energy. Although their theory concerns the dynamical-type Jahn-Teller effect, similar discussion in terms of the effective interactions could be given for the present static-JT case.

Of course this scenario is just a speculation and so different interpretations are, if possible, welcomed. Anyway, convincing discussions about the origin of the general stability of the disproportionation state is essential. Fortunately, the disproportionation state is uncorrelated according to the authors; it should allow simple explanations (e.g., discussion in terms of effective interactions based on the mean-field picture).

Our response:

The "negative J " model, simulating the JT effect in fullerides, was introduced by Capone, Tosatti and Fabrizio to incorporate the JT effect in heavy DMFT calculations. It is a very simplified treatment by far not accounting the physical richness of the JT effect in C_{60}^{n-} anions. Fortunately we do not need to apply this artificial description here, neither for performing the

calculations nor for the explanation of the obtained results, which are rationalized in terms of the competition of the band energy and orbital-splitting energy (arising from JT effect and the intrinsic crystal-field splitting in non-cubic lattices) as described above.

Reviewer #2

Comment #1:

The present manuscript describes the results of theoretical calculations of the electronic phase diagram of A_4C_{60} , an alkali-doped fulleride with an even doping number. The results presented here are an extension of the methodology developed by the same authors and applied to the metallic and insulating phases of A_3C_{60} . The authors' work revolves around the importance of the dynamic/static Jahn-Teller effect in contributing towards the rationalization of the electronic properties of fulleride solids. Overall I find the ideas and results here of interest but the manuscript is written in a somewhat opaque and cryptic way making it hard to follow the development of the results. Definitely it should be re-written in a more logical and coherent way to allow the reader to follow the results.

Our response:

In the revised version of the manuscript we rewrote significantly the text and added several new paragraphs explaining in detail the meaning of the obtained results, especially, the origin and stabilization of the two competing phases, the homogeneous correlated metallic phase and the non-correlated band-insulating phase.

Comment #2:

The major message of this work is the universality of orbital disproportionation in fulleride physics. But unlike A_3C_{60} which is cubic (face-centered or body-centered-derived), all known A_4C_{60} fullerides suffer from a reduction of symmetry to either tetragonal (K_4C_{60} , Rb_4C_{60}) or orthorhombic (Cs_4C_{60}). It is unclear to the present referee what the role of orbital disproportionation is here when the degeneracy of the t_{1u} orbitals is lifted. In the case of Cs_4C_{60} (which is not considered by the authors) there is full lifting of the degeneracy.

Our response:

The Reviewer correctly states that the orbitally disproportionated phase is stabilized by both JT distortions and intrinsic crystal-field splitting (inherent to non-cubic lattices) of the t_{1u} LUMO orbitals on the fullerene sites. However, the latter alone cannot be responsible for the band-insulating solution, therefore, the JT distortions on C_{60} are indispensable for its stabilization in A_4C_{60} fullerides. Actually the intrinsic crystal-field splitting is only relevant in the Cs_4C_{60} , with

the bco lattice, and has no influence whatsoever on the stabilization of the orbitally disproportionated (band insulating) phase in K_4C_{60} and Rb_4C_{60} fullerenes with the bct lattice. This is clear from the criterion in Table 1, which is discussed at length in the present version of the manuscript. In the revised version, we consider the case of Cs_4C_{60} too, adding the corresponding crystal-field splitting scenario to Table 1.

Comment #3:

The authors begin by using a hypothetical cubic fcc structure for A_4C_{60} which could be a starting model of course. They do say that this does not affect the main conclusions but I really cannot see that this is developed in the manuscript. There is Table I where the last line should correspond to "real" materials. Here the crystal field has already partially removed the degeneracy but its magnitude does not enter the criteria definitions.

Our response:

As answered to the previous comment, the crystal-field splitting of the t_{1u} LUMO orbitals, intrinsic to non-cubic lattices of A_4C_{60} , cannot lead alone to the orbitally disproportionated phase and to the observed band insulating state. Therefore, for the sake of simplicity, we started by considering a hypothetical cubic fcc structure for A_4C_{60} , which is sufficient for a full qualitative description of the discussed effects. That is the reason for us to say that using such a cubic lattice for A_4C_{60} does not affect the main conclusions of the work. We then considered the real lattices for A_4C_{60} (bct and bco) and the effect of the crystal field produced by them on the criterion of stabilization band insulating phase (Table 1). As mentioned in the previous comment and in the revised text of the manuscript, the crystal-field splitting does not enter the criterion for the band insulating state of fullerenes with bct lattice (K_4C_{60} and Rb_4C_{60}) but influence this criterion in the case of fullerenes with bco lattice, such as Cs_4C_{60} .

Comment #4:

For the parameters used, the material should be always a band insulator rendering the conclusions entirely trivial. Moreover body-centered orthorhombic Cs_4C_{60} will have three non-degenerate states rendering the relevance of the JT effect questionable.

Our response:

This is absolutely not the case, which is actually one of the messages of our work. The intrinsic crystal-field splitting of the t_{1u} LUMO orbitals on C_{60} sites in fullerenes does not render them automatically band insulators. Figures 5A and 5C show the t_{1u} LUMO band of K_4C_{60} calculated by DFT. We can see that it is not a band insulator but rather a metal despite the intrinsic crystal-field splitting of 130 meV (extracted from DFT). The same situation is realized in Cs_4C_{60} and any other fullerene in which the intrinsic crystal-field splitting is significantly smaller than the uncorrelated bandwidth. The band insulating state only arises due to Jahn-Teller distortions on

sites and in the presence of electron repulsion in the t_{1u} shell (Hubbard U), as Figures 5B and 5D show.

Comment #5:

I would really like to see the treatment by the authors presented in a coherent way. One can start by the fcc model but if one wants to address real systems one should emphasize the treatment of the bct and bco structures.

Our response:

We hope to have achieved this in the revised version of the manuscript, in which we indeed discuss the effect of fcc, bct and bco lattices on the band insulating state.

Comment #6:

While the splitting of the levels in experimental bct (because $c < a$) is like shown in Fig 5c inset, most of the other figures deal with inverted case relevant to $a < c$ (e.g. Fig 4).

Our response:

In Fig. 4, we discuss the JT splitting in the fcc lattice and not the crystal-field splitting of bct lattice.

Comment #7:

In addition, various parameters appear without much explanation, e.g. we can see Δ , Δ_0 , Δ_c .

Our response:

Δ_0 is crystal-field splitting, Δ is the sum of crystal-field and Jahn-Teller splittings, and Δ_c is the critical Δ for which the correlated metal to band insulator transition realizes.

In the revised manuscript, we clearly describe all these parameters.

Comment #8:

It is not clear to me what the authors imply by saying that A_4C_{60} are uncorrelated band-insulators despite the large on-site repulsions. It looks like one is playing with words here. What is the physical significance of such statements.

Our response:

The uncorrelated band insulating state is described by single Slater determinant, whereas the correlated state (e.g., correlated Gutzwiller's wave function) is by definition multi-determinant

one. We are not playing with the words here: the increasing of electron correlation in the t_{1u} orbitals of C_{60} sites (increasing the parameter U) leads indeed to a transition from a correlated metal state to the uncorrelated band-insulating state. The last is obtained as an exact solution of the Hamiltonian (1), which we discuss in the text and in the ESI.

Comment #9:

I find interesting the statement that the picture developed before mainly by the Tosatti/Gunnarsson groups of the importance of degeneracy in stabilizing the highly-correlated metallic state in A_3C_{60} is inappropriate. To this effect they stress the importance of the link between the JT effect and the the lifting of the degeneracy. This is a very interesting issue. On the other hand, it will be fair to also refer to theoretical work which does not include any phonon contributions, notably the recent work by Kivelson which finds that a low-spin ground state (ie orbital degeneracy lifted) can occur purely on electronic grounds in fullerides (PRB 93, 165406 (2016)). All these are interesting ideas in this exciting field and one needs to keep an open mind.

Our response:

The recent work by Jiang and Kivelson (PRB 93, 165406 (2016)) addressed the pairing mechanism in A_3C_{60} fullerides by considering the electronic structure of single C_{60} anions within the t - J model. We do not consider this work as being relevant to real fullerene and fullerides because (i) the t - J model is not suitable for the description of the electronic structure of organic molecules, and (ii) the obtained results contradict the known experimental facts. (i) In a single C_{60} molecule, the effective intra atomic electronic repulsion on the carbon atoms (parameter U) is relatively weak compared to the spread of the π molecular orbitals, while the criterion of validity of the t - J model supposes the opposite, a very large (infinite) U . (ii) The result obtained by the authors of the mentioned work is found in contradiction to the experimental observation of an antiferromagnetic Mott insulating phase in Cs_3C_{60} . They obtained the intersite charge disproportionated configuration $C_{60}^{2-} + C_{60}^{4-}$ as more stable state than two C_{60}^{3-} , whereas the former state is non magnetic and, therefore, cannot be associated with the magnetic ground state of Cs_3C_{60} .

Pure electronic mechanisms of the sort considered by Jiang and Kivelson have been proposed long time ago by Varma, Kivelson and Zaanen (see, e.g., their paper in Science, 1990) and have been criticized latter as contradicting the real situation in fullerides (see, e.g. the review by Gunnarsson in RMP, 1997).

Comment #10:

I think this paper is possibly publishable but the authors should improve its readability considerably and give a more convincing explanation of "orbital disproportionation" for systems which are never cubic (cf. A_3C_{60}) and for which in reality there is always a strong competing crystal field.

Our response:

We hope of having achieved the clarity of the presentation in the revised version of the manuscript.

Reviewer #3

Comment #1:

In this work the authors present a rather detailed analysis of a quite cumbersome model, containing several phenomenological parameters to describe both the electron correlation and the Jahn-Teller effect.

Our response:

Our model is not phenomenological but microscopic, involving realistic orbital vibronic coupling constant g , frequency ω , transfer parameters and the Hund's rule coupling parameter. In Ref. 26, we have simulated the high-resolution photoelectron spectrum of C_{60}^- and derived the orbital vibronic coupling constants for all active vibrational modes, which agree well with the calculated DFT values. With the same DFT approach, we calculated the Hund's rule coupling parameter (Ref. 29). The constants g and ω for the effective mode describing the JT effect on the fullerene sites were determined by projecting the low-lying vibronic levels of C_{60}^{3-} (Ref. 29). The only free parameter of the theory is the Coulomb repulsion in the t_{1u} orbitals of C_{60} sites (U), for which we expect a range of 0.4-0.7 eV based on available calculations and experimental data.

Comment #2:

The main effect is that the Coulomb repulsion, since provides an effective repulsion between empty and occupied states, can amplify the Jahn-Teller distortion and provide a change in the occupied states, leading to band insulator behavior. This effect is qualitatively reproduced within the simple Hartree-Fock theory (see e.g. PRB 62, 7619) that, as expected, overestimates the gap. The present work represents an improvement of previous works, since the Gutzwiller approximation may be more quantitative than the simplest Hartree-Fock theory. However I do not think this paper deserves to be published in this Journal because of lack of importance and novelty of the results presented.

Our response:

In our previous work the band insulating state was supposed, not derived. In the present work we derive this state as a ground electronic phase in fullerides, basing on a realistic Hamiltonian and an adequate treatment of electron correlation and vibronic interactions. This proof of the band

insulating ground state was not given in the previous work. Actually, the community of researchers working in the field of fullerenes still believe that the insulating state in A_4C_{60} is of Mott-Hubbard origin – in all publications related to A_4C_{60} these fullerenes are termed as “Mott-Jahn-Teller” insulators. Therefore, the proof that A_4C_{60} (and A_2C_{60}) are uncorrelated band insulators is a major advance in the field, contrary to the statement of the Reviewer. Second we prove here that there exist two competing electronic phases, the correlated metallic and the band insulating, which are stabilized in function of the strength of the electron repulsion on C_{60} sites (U).

Comment #3:

Moreover from the experimental point of view, if a critical value of U is necessary for a Jahn-Teller orbital disproportionation of electronic density, this should be a measurable effect (e.g. by increasing the temperature). However I do not see any evidence from the experimental data, and the model and approximations used, certainly require some experimental confirmation.

Our response:

The existence of the critical U is consistent with the observation of the metal-insulator transition in Rb_4C_{60} under pressure (Ref. 43). Under pressure, the electron transfer is enhanced and U becomes weaker due to the screening. Thus, with sufficiently large pressure, U becomes smaller than the critical value and the system becomes a metal in accord with experiment.

Reviewers' comments:

Reviewer #1 (Remarks to the Author):

The authors sincerely considered my comments. In the previous round I requested the authors to add convincing interpretation about why their interesting ground state is numerically stabilized. I have put a speculation that the mechanism of the disproportionation could be related to that in a preceding theory of negative J by Capone et al. The authors, in contrast to my speculation, stated that the present mechanism is not related to the Capone's one. Reading the authors' description, however, I am rather coming to feel that my speculation is correct: They are closely related, at least for the A3C60 cases. Accordingly, I am also coming to an idea that the Iwahara-Chibotaru description may not be perfectly applicable to the A3C60 case, which could reduce the universality of the authors' theory. I append below my understanding of the two theories.

The commonality is represented by the following two points: i) The coupling of electrons to lattice degree of freedom is the fundamental driving force of the disproportionation [please note that the Capone's negative J has been quantitatively attributed by Nomura et al. to the phonon-mediated J -type electron-electron interaction (Sci. Adv. 1, e1500568 (2015); PRB 92, 245108 (2015))], ii) The suppression of the kinetic energy (=bandwidth) by the electron-electron Coulomb interaction helps the energy gain of the disproportionation to dominate (see right column of pp. 3 of Nomura et al., Sci. Adv.). The differences are the following: I) the Capone-Nomura theory treats the lattice degree of freedom as the phonons defined around the undistorted configuration, whereas the Iwahara-Chibotaru theory treats that as a classical parameter q . II) The former and latter implicitly considers the antiadiabatic and adiabatic limit, respectively. III) The counterparts of the bandwidth in the respective theories are the coupling to quenched (but dynamically oscillating) and unquenched (but possibly non-directional in the time average) distortions.

The two theories seem to be somehow capturing the right physics in different regimes and have their own drawbacks. Of course in the case of isolated molecule, the latter theory is absolutely more appropriate. However, in the solid case, the former is also appropriate to some extent since the potential energy surface has minimum at the undistorted configuration and therefore the Jahn-Teller phonon is well-defined. Also, with the previously reported experimental signatures of the dynamical Jahn-Teller effect, probably one cannot judge which theory is correct. For example, the low-temperature separation of the IR absorbance peaks (Klupp et al., Nat. Commun. 3, 912 (2012)) will be reproducible with the both theory; the Capone-Nomura theory gives dynamically disproportionated state (see Fig.3C of Nomura et al., Sci. Adv.), which obviously affect the phonon degeneracy. In addition, the Iwahara-Chibotaru theory seems to have a drawback that it cannot be straightforwardly extended for describing the superconducting state, despite they have addressed the metallic phase in the vicinity of the Mott insulating phase [Iwahara and Chibotaru, PRB 91, 035109 (2015)]. With this fact I am coming to regard that the "universality" of their description is a bit overstatement.

In summary, I can accept that the disproportionation phenomenon is indeed a universal feature of A_nC₆₀, but cannot that the present theory is wholly the most appropriate. Another theory, which is probably more appropriate in the different regime, can also reproduce the dynamical disproportionation for A₃C₆₀. Since the authors' main statement is about the universality of the phenomenon (not of the theory), preceding theories reproducing that phenomenon should be discussed anywhere in the text. With the proper relation of the present work with other related works, I can recommend its publication.

An interesting possibility arises. If the authors can conclude with a convincing discussion that the apparent success of the Capone-Nomura theory for A₃C₆₀ (especially in the small-volume regime) is an artifact and the Iwahara-Chibotaru theory dominates, the universality in a stronger sense is established. This could further enhance the value of the manuscript, bringing it above the average of the NCOMM articles.

Reviewer #2 (Remarks to the Author):

The authors took into account seriously the reviewers' comments and queries/criticisms. Although I still remain not entirely clear about the authors' proposals, I recommend publication as a useful addition to the literature on these complex molecular systems that should generate extensive discussion.

Reviewer #3 (Remarks to the Author):

Though I am not fully satisfied by the authors reply and the related modifications to the manuscript, I believe the manuscript can be now considered for publication in this Journal.

Response to the reviewers' comments

Reviewer #1

Comment #1:

The authors sincerely considered my comments. In the previous round I requested the authors to add convincing interpretation about why their interesting ground state is numerically stabilized. I have put a speculation that the mechanism of the disproportionation could be related to that in a preceding theory of negative J by Capone et al. The authors, in contrast to my speculation, stated that the present mechanism is not related to the Capone's one. Reading the authors' description, however, I am rather coming to feel that my speculation is correct: They are closely related, at least for the A_3C_{60} cases. Accordingly, I am also coming to an idea that the Iwahara-Chibotaru description may not be perfectly applicable to the A_3C_{60} case, which could reduce the universality of the authors' theory.

Our response:

The two approaches, the ours and the Capone's "negative- J " model, are related since they both treat concomitantly the effect of electron correlation and Jahn-Teller effect in fullerides. However, they differ essentially in the treatment of the Jahn-Teller effect on C_{60} sites. While our description of Jahn-Teller effect on the fullerene sites is exact, the Capone's approach replaces the electron-vibrational interaction on fullerene sites with an effective electronic operator of the Hund form with negative constant J . Because of these differences in our approaches, the obtained results obviously differ as well. Given that our approach does not use any approximations for the description of Jahn-Teller effect, we are convinced that the results obtained in the present manuscript (the orbital disproportionation of the electronic density) are correct.

Concerning the A_3C_{60} , these fullerides do display orbital disproportionation as well, contrary to Reviewer's statement. This was demonstrated in our previous article, Phys. Rev. B 91, 035109, 2015 (Ref. [30] of the present manuscript). The similar behavior of A_3C_{60} and A_4C_{60} as function of U , showing a transition to orbitally disproportionated phase at some critical value U_c , can be inferred from Figure 2 of the present manuscript. It is precisely this similar behavior of the two kind of fullerides which allows us to draw the conclusion of the universality of orbital disproportionation of electronic density in fullerides.

Comment #2:

I append below my understanding of the two theories.

The commonality is represented by the following two points: i) The coupling of electrons to

lattice degree of freedom is the fundamental driving force of the disproportionation [please note that the Capone's negative J has been quantitatively attributed by Nomura et al. to the phonon-mediated J -type electron-electron interaction (Sci. Adv. 1, e1500568 (2015); PRB 92, 245108 (2015))], ii) The suppression of the kinetic energy (= bandwidth) by the electron-electron Coulomb interaction helps the energy gain of the disproportionation to dominate (see right column of p. 3 of Nomura et al., Sci. Adv.). The differences are the following: I) the Capone-Nomura theory treats the lattice degree of freedom as the phonons defined around the undistorted configuration, whereas the Iwahara-Chibotaru theory treats that as a classical parameter q . II) The former and latter implicitly considers the antiadiabatic and adiabatic limit, respectively. III) The counterparts of the bandwidth in the respective theories are the coupling to quenched (but dynamically oscillating) and unquenched (but possibly non-directional in the time average) distortions.

Our response:

Although the “negative- J ” model is indeed obtained from the conventional Jahn-Teller Hamiltonian by integrating out the phonons, this does not imply whatsoever its equivalence to the latter. In fact the “negative- J ” model is oversimplified, since a complex vibronic coupling of the three LUMO orbitals to several tens on nuclear modes on each fullerene site cannot be simulated adequately by pure electronic operator.

What is written in the right column of p. 3 of Nomura et al., Sci. Adv. is not at all a disproportionated phase but just a homogeneous electronic phase. Indeed, in their Monte Carlo simulation they obtained a phase in which six different orbital configurations with equal weight coexist (Fig. 4C of their paper). This implies equal population of all three LUMO orbital components on each fullerene site ($\langle n_x \rangle = \langle n_y \rangle = \langle n_z \rangle$), i.e. the lack of orbital disproportionation.

The existence of Jahn-Teller effect on fullerene sites, which is the ultimate reason for the orbital disproportionation of the electronic density, requires the distortions (static or dynamic) on each fullerene site. The “phonons defined around the undistorted configuration” cannot describe the Jahn-Teller effect in principle – this approach is only applicable to crystals with orbitally nondegenerate sites, where the nuclear dynamics reduces to small harmonic vibrations around one single equilibrium point. In fullerene anions C_{60}^{3-} there is no one single minimum (equilibrium point) but a three-dimensional continuum of minima after active Jahn-Teller distortions.

In our treatment the nuclear coordinate q is not a “classical parameter” like states the Reviewer but a true quantum variable. In our approach we treat the vibronic coupling exactly, via the solution of the corresponding Schrödinger equation for the dynamical Jahn-Teller effect on sites. On the other hand the orbital disproportionation effect does not depend crucially on whether the Jahn-Teller effect on fullerene sites is static or dynamic. The dynamic character of Jahn-Teller distortions only enhances the tendency towards orbital disproportionation as we have demonstrated for A_3C_{60} in our previous work Phys. Rev. B 91, 035109, 2015 (Ref. [30] of the present manuscript) and confirm here again for A_4C_{60} (see Table I of the present manuscript).

Comment #3:

The two theories seem to be somehow capturing the right physics in different regimes and have their own drawbacks. Of course in the case of isolated molecule, the latter theory is absolutely more appropriate. However, in the solid case, the former is also appropriate to some extent since the potential energy surface has minimum at the undistorted configuration and therefore the Jahn-Teller phonon is well-defined. Also, with the previously reported experimental signatures of the dynamical Jahn-Teller effect, probably one cannot judge which theory is correct. For example, the low-temperature separation of the IR absorbance peaks (Klupp et al., Nat. Commun. 3, 912 (2012)) will be reproducible with the both theory; the Capone-Nomura theory gives dynamically disproportionated state (see Fig.3C of Nomura et al., Sci. Adv.), which obviously affect the phonon degeneracy. In addition, the Iwahara-Chibotaru theory seems to have a drawback that it cannot be straightforwardly extended for describing the superconducting state, despite they have addressed the metallic phase in the vicinity of the Mott insulating phase [Iwahara and Chibotaru, PRB 91, 035109 (2015)]. With this fact I am coming to regard that the “universality” of their description is a bit overstatement.

Our response:

We emphasize once again, that the two approaches, the ours and the Capone’s, treat the same systems. The only difference between them is that we do not apply any approximation for the treatment of Jahn-Teller interaction on fullerene sites, while Capone simulate it by an effective electronic operator. Therefore, one cannot state that our approach has a drawback in some “regime” compared to the Capone’s approach. We also do not understand the Reviewer’s statement why the true electron vibrational operator, used in our approach, is not applicable to a solid (?). We did apply already this Hamiltonian with static and dynamic Jahn-Teller effect on sites in our previous treatment of A_3C_{60} , Phys. Rev. B 91, 035109, 2015 (Ref. [30] of the present manuscript).

We do not agree with the Reviewer that both approaches can describe equally well the IR absorbance spectra of Cs_3C_{60} (Klupp et al., Nat. Commun. 3, 912 (2012)). The IR spectra describe the transition between vibrational levels, while in the Capone’s approach the vibrational degrees of freedom are already projected out.

We also do not agree with the Reviewer’s statement that the Capone-Nomura theory gives dynamically disproportionated state. The latter necessarily implies the existence of dynamical Jahn-Teller effect on fullerene sites. The corresponding wave function includes the adiabatic electronic wave function, involving Jahn-Teller nuclear modes, and the dynamical wavefunction after these modes. A net electronic wave function like in Capone’s approach cannot describe the dynamical Jahn-Teller effect and the dynamical disproportionation in principle. An example of description of dynamical disproportionation is given in our previous work Phys. Rev. B 91, 035109, 2015 (Ref. [30] of the present manuscript) on A_3C_{60} fullerenes.

We do not understand the statement of the Reviewer that our approach cannot be straightforwardly extended for the description of the superconducting state. There are plenty examples of straightforward application of true electron-phonon Hamiltonians to the description of superconductivity. These are based on the routine use of Eliashberg theory of superconductivity and we do not see any restrictions to apply it to fullerides.

Comment #4:

In summary, I can accept that the disproportionation phenomenon is indeed a universal feature of A_nC_{60} , but cannot that the present theory is wholly the most appropriate. Another theory, which is probably more appropriate in the different regime, can also reproduce the dynamical disproportionation for A_3C_{60} . Since the authors' main statement is about the universality of the phenomenon (not of the theory), preceding theories reproducing that phenomenon should be discussed anywhere in the text. With the proper relation of the present work with other related works, I can recommend its publication.

Our response:

As we already stated, there is no “another theory” describing disproportionation in fullerides. The works by Capone and Nomura use pure electronic Hamiltonian and, therefore are unable to describe the orbital disproportionation in fullerides. Indeed, the solution for the electronic phase in the article of Nomura et al., Sci. Adv. merely describes a homogeneous electronic state, with all orbitals equally populated (see our response to the comment#2), but not an orbitally disproportionated state. We also do not know works from other groups which proved the existence of disproportionated ground state in fullerides.

Comment #5:

An interesting possibility arises. If the authors can conclude with a convincing discussion that the apparent success of the Capone-Nomura theory for A_3C_{60} (especially in the small-volume regime) is an artifact and the Iwahara-Chibotaru theory dominates, the universality in a stronger sense is established. This could further enhance the value of the manuscript, bringing it above the average of the NCOMM articles.

Our response:

Actually the theory of Capone-Nomura treats the superconductivity in the large-volume regime and is apparently not applicable to small-volume regime (cf. Fig.2A and Fig.2B of their article). In the small-volume regime their prediction will be opposite to the experimental observation. That is, the predicted superconducting T_c will increase with the reduction of the volume of A_3C_{60} in contrast to the experimental observation (Fig. 2B). Indeed, this erroneous trend was found in a very similar treatment of superconductivity by J.E. Han, O. Gunnarsson and V.H. Crespi, Phys. Rev. Lett. 90, 167006 (2003) (see Fig. 4 in that publication).

Concerning the orbital disproportionation in fullerides, it cannot be described by the Capone-Nomura theory as we commented above. Moreover, these authors never stated in their publications that the solution for the electronic phase obtained by their “negative- J ” model describes orbitally disproportionated LUMO electronic density in fullerides. Therefore, the Reviewer can be ensured that our work is brand original, with sufficient degree of novelty to be considered for publication in Nature Communications.

Reviewer #2:

Comment:

The authors took into account seriously the reviewers' comments and queries/criticisms. Although I still remain not entirely clear about the authors' proposals, I recommend publication as a useful addition to the literature on these complex molecular systems that should generate extensive discussion.

Our response:

We thank the Reviewer for the appreciation of our work. We are convinced that our work contains sufficient novelty to be of high interest for a broad community of researchers.

Reviewer #3:

Comment:

Though I am not fully satisfied by the authors reply and the related modifications to the manuscript, I believe the manuscript can be now considered for publication in this Journal.

Our response:

We thank the Reviewer for his trust of the importance of our work and his recommendation for publication.

List of changes

In the revised manuscript, we added two sentences on the origin of the disproportionation at the end of Section III.

REVIEWERS' COMMENTS:

Reviewer #1 (Remarks to the Author):

Actually I did not feel convinced with the authors' rebuttal. However, I have at least found a crucial misunderstanding behind my opinion about the authors' previous work for A3C60: if the distortion q is treated classically or quantum mechanically. With the renewed knowledge my view on the authors' theory has much changed.

Although there are remaining issues for me, to accelerate publication, I admit the closing of my review. Let me append below the issues I have not yet been convinced, which do not have to be responded in the present manuscript.

1,

What I called "disproportionation" in the Capone-Nomura theory is the quantitative growth of the fraction of the (2 1 0)-weighted state around volume $> 760 \text{ \AA}^3$ in Fig. 4C. It is true that $\langle n_x \rangle = \langle n_y \rangle = \langle n_z \rangle$ even in that case, but such phase is undoubtedly different from the trivial metallic phase, where all the configurations including the (1 1 1) state have nearly equal weight. Therefore I was not convinced with the following rebuttal of the authors: "Indeed, in their Monte Carlo . . . i.e. the lack of orbital disproportionation."

2,

I am not yet convinced that the phonon description starting from the undistorted configuration should be abandoned. For the solid fullerene (NOT isolated C_{60}^{-3}) it is obvious that the adiabatic potential surface has its global minimum at the undistorted configuration in a certain regime. Even in the case of the authors' calculation, when $U_{\parallel} \sim 0.62$, the adiabatic energy surface has its global minimum at $q=0$ (Fig.1A in the main text), where I think that the phonon theory is well-defined. I view whether the experimental situation corresponds to $U_{\parallel} \sim 0.62$ or > 0.67 is in principle a sensitive issue, though I agree with the trend--stronger U_{\parallel} should make the $q \neq 0$ configuration the ground state.

Response to the Reviewer's comments

Reviewer #1

Comment #1:

Actually I did not feel convinced with the authors' rebuttal. However, I have at least found a crucial misunderstanding behind my opinion about the authors' previous work for A_3C_{60} : if the distortion q is treated classically or quantum mechanically. With the renewed knowledge my view on the authors' theory has much changed.

Although there are remaining issues for me, to accelerate publication, I admit the closing of my review. Let me append below the issues I have not yet been convinced, which do not have to be responded in the present manuscript.

Our Response:

We are grateful to the Reviewer for his comment and a positive attitude to our work.

Comment #2:

1,

What I called "disproportionation" in the Capone-Nomura theory is the quantitative growth of the fraction of the (2 1 0)-weighted state around volume $> 760 \text{ \AA}^3$ in Fig. 4C. It is true that $\langle n_x \rangle = \langle n_y \rangle = \langle n_z \rangle$ even in that case, but such phase is undoubtedly different from the trivial metallic phase, where all the configurations including the (1 1 1) state have nearly equal weight. Therefore I was not convinced with the following rebuttal of the authors: "Indeed, in their Monte Carlo . . . i.e. the lack of orbital disproportionation."

Our Response:

Of course the state with suppressed (1 1 1) and six dominant configurations of (2 1 0) type is different from a trivial metallic state with similar weights for all orbital configurations. However, even if only (2 1 0) states are obtained as dominant in the Capone-Nomura theory, the fact that there are six equally weighted configurations of this type will quench the disproportionation because $\langle n_x \rangle$, $\langle n_y \rangle$, and $\langle n_z \rangle$ are equal anyway.

Comment #3:

2,

I am not yet convinced that the phonon description starting from the undistorted configuration should be abandoned. For the solid fullerene (NOT isolated C_{60}^{-3}) it is obvious that the adiabatic potential surface has its global minimum at the undistorted configuration in a certain regime. Even in the case of the authors' calculation, when $U_{\parallel} \sim < 0.62$, the adiabatic energy surface has its global minimum at $q = 0$ (Fig. 1A in the main text), where I think that the phonon theory is well-defined. I view whether the experimental situation corresponds to $U_{\parallel} \sim < 0.62$ or > 0.67 is in principle a sensitive issue, though I agree with the trend--stronger U_{\parallel} should make the $q \neq 0$ configuration the ground state.

Our Response:

The phonon description, of course, should not be abandoned. Indeed, as the Reviewers write, it is just the same as in conventional crystals for $U_{\parallel} \sim < 0.62$, the adiabatic energy surface has its global minimum at $q = 0$. The knowledge of exact value of U_{\parallel} is therefore crucial. However, from all available estimates, it is larger than 0.6 eV (we discuss this issue at more length in our previous paper, Ref. [30]). Therefore, we are convinced that the (dynamic) Jahn-Teller instability takes place in all fullerenes,

which allows us to state the universality of the orbital disproportionation in these materials.